# Five-Dimensional Optical Data Storage Based on Ellipse Orientation and Fluorescence Intensity in a Silver-Sensitized Commercial Glass

**DOI:** 10.3390/mi11121026

**Published:** 2020-11-24

**Authors:** Chang-Hyun Park, Yannick Petit, Lionel Canioni, Seung-Han Park

**Affiliations:** 1Department of Physics, Yonsei University, Seoul 03722, Korea; changhyun.park@u-bordeaux.fr; 2University of Bordeaux, CNRS, CEA, CELIA, UMR 5107, 351 Cours de la Libération, 33405 Talence CEDEX, France; lionel.canioni@u-bordeaux.fr; 3University of Bordeaux, CNRS, ICMCB, UMR 5026, 87 Avenue du Dr. A. Schweitzer, 33608 Pessac, France

**Keywords:** multi-dimensional optical data storage, direct laser writing, phase shaping, femtosecond phenomena, specialty glasses, silver clusters

## Abstract

Five-dimensional (5D) recording and decoding is demonstrated by using femtosecond direct laser writing in a silver-containing commercial glass. In particular, laser intensities and ellipse orientations generated by anamorphic focusing are employed to produce 5D data storage unit (3D for XYZ, 1D for the orientation of the elliptically-shaped data storage unit and 1D for its fluorescence intensity). In the recording process, two different images of a 4-bit bitmap format were simultaneously embedded in the medium by multiplexing the elliptical orientation of the laser focus and its intensity so as to access oriented elliptical patterns with independent fluorescence intensity. In the decoding process, two merged original images were successfully reconstructed by comparing each data storage unit with a fabricated calibration matrix of 16 × 16 levels for elliptic orientations and fluorescence intensities. We believe this technique can be applied to semi-permanent high-density data storage device.

## 1. Introduction

High-density optical data storage (ODS) techniques have been actively investigated to overcome storage-capacity and life-time limitations of conventional storage media such as CDs and DVDs [1,2]. Multilayered 3D data storage was demonstrated by using two-photon excitation in photopolymer [3,4]. The polarization–multiplexed optical memory technique in urethane–urea copolymers was also presented [5]. To increase the data storage capacity, 5D optical recording was proposed by using three different wavelengths and two polarizations of an incident laser [6]. Pearson et. al. fabricated bow-tie nanoscale fuses on graphene for permanent write-once-read-many applications [7]. Zhang et al. achieved semi-permanent 5D optical recording and decoding of digital information by femtosecond (fs) laser nanostructuring in fused quartz [8,9]. Recently, polychromatic and polarized multilevel multiplexing in glass containing gold nanorods (GNRs) was also demonstrated [10].

In general, nano and micro-scale ODS units created in glass (mostly fused silica) result from type I or type II index modifications produced by direct laser writing (DLW) [11,12], corresponding either to an isotropic refractive index change or to the creation of self-organized form-birefringent nano-gratings, respectively. High laser intensities are required for both of these DLW techniques. In contrast, the recently reported type Argentum (type A) index modification can be performed in silver-containing glass with relatively low incident laser irradiation around 10 TW/cm^2^ [13], taking advantage of tailored materials with sensitizing agents [14]. Type A DLW of fluorescent hollow cylindrical structures was demonstrated, which has been intensively investigated in terms of spatial 3D distributions of associated linear (absorption, excitation, emission, and lifetime, index modification) and nonlinear (SHG and THG) micro-spectrocopies, as well as in terms of silver element redistribution (electron chemical microprobe, near-field optical scanning microscopy) and of detailed numerical modeling under many-pulse laser-glass interactions, as reviewed by Petit et al. [15,16]. Indeed, these fluorescent hollow cylindrical structures show an increasing readout fluorescence emission while DLW intensity increases [15]. An accelerated aging test, carried out in a climatic test chamber at 100 °C for 3100 h, revealed long-term storage capability of this type A approach, with unmodified optical performance and fluorescence spectrum changes less than 1% [17]. The three-layer images, inscribed by from Royon et al. ten years ago, are still perfectly observable [18].

In this letter, we report on the 5D ODS technique based on fluorescence intensity and ellipse orientation of data storage units by using Type A DLW in silver-containing commercial glass. Fluorescence intensity modulation, obtained by controlling the incident laser irradiance, was used to provide one dimension to ODS, in addition to the three spatial coordinates. In order to increase the experimental degrees of freedom, the hollow cylindrical structure was transformed into a hollow ellipse structure by anamorphic focusing of the fs laser. We demonstrate that two different 4-bit bitmap images of 100 × 100 pixels can be embedded into one single image in such glasses by multiplexing 16 levels of the laser intensity and 16 levels of the ellipse orientation. We also show that the original images could be successively decoded and retrieved.

## 2. Materials and Methods

### 2.1. Commercial Glass Sample

The sample used in our 5D ODS experiments was a silver-containing commercial glass (AG01^®^, from ArgoLight Company, Pessac, France). Fluorescent clusters were created in AG01^®^ with a near-infrared fs laser, enabling the type A DLW. As detailed elsewhere, silver-sensitized glasses and laser-activated photochemistry can lead to the fabrication of innovative integrated or surface waveguides, directional couplers both in bulk or ribbon fiber-shaped samples [15]. In the present work, laser inscription were conducted in fixed point-by-point positions, leading to localized patterns with controlled orientations and fluorescence intensities, as detailed hereafter.

### 2.2. Formation and Observation of Oriented Elliptical Fluorescent Patterns

Figure 1a shows our experimental setup for recording 4-bit bitmap images. The light source for Type A DLW was an Yb:KGW fs laser (Amplitude system, 9.8 MHz repetition rate, 390 fs pulse FWHM duration and operating at 1030 nm). An Acousto-Optic Modulator (AOM) was used to generate the 16-step incident laser intensities. The phase of the laser beam was adjusted by a liquid-crystal-silicon spatial light modulator (SLM; LCOS, X10468-03, 800 × 600 pixel, Hamamatsu Photonics, Higashi-ku, Japan). The beam was focused by an objective lens (Zeiss, Oberkochen, Germany, 20×, 0.75 NA).

Laser-induced fluorescent patterns, composed of fluorescent molecular silver clusters, were imaged with a scanning confocal microscope (Leica, Wetzlar, Germany, DM6 CFS TCS SP8) with 1 Airy disk pinhole (defined for a central wavelength of 500 nm) and a Leica microscope objective (20×, NA = 0.75), integrating the 450–550 nm spectral fluorescence emission under excitation with a 405 nm laser. The limited UV illumination time of each pattern during the scanning imaging process did lead to any fluorescence intensity decay nor to any permanent fluorescence bleaching.

Figure 1b shows the SLM holographic phase masks with a circular profile (left) and a cylindrical profile (right). Indeed, we introduce an SLM phase mask with the cylindrical profile to generate an elliptical fluorescent pattern at the tangential focus, to control the orientation encoding direction [19]. The 16-step orientations of the elliptical pattern at the tangential focus were created by gradually rotating the SLM mask while maintaining the cylindrical profile.

Figure 1c shows the 16-step orientation direction of the fluorescence patterns produced by Type A DLW from 0° to 168.75° with the step of 11.25°. Astigmatism was intentionally introduced by anamorphic focusing [20]. Fluorescence intensity at the tangential focus was observed to be the strongest with a clear oval pattern. The eccentricity of the ellipse was also adjusted to be the best for retrieving the orientation direction after DLW.

### 2.3. Fluorescence Calibration Matrix with 16 Intensity Levels and 16 Ellipse Orientations

DLW’s laser intensity was tuned by a AOM so that the incident intensity at the focal point was varied from 9.4 TW/cm^2^ to 12.7 TW/cm^2^ (pulse energy of 41.7~56.2 nJ). Since laser intensity is not perfectly linear with the AOM voltage, applied voltages were carefully adjusted to become evenly spaced laser intensity for the 16-step DLW, leading to the 2^4^ = 16 intensity-encoded levels. The lowest DLW intensity was selected slightly above the intensity threshold for the creation of identifiable elliptical patterns. It should be noted that the range of laser intensities was carefully chosen to simultaneously allow for the dual readout of both the ellipse orientation and fluorescence intensity.

Figure 2a shows the 2^8^ = 256 fluorescence matrix fabricated by 2^4^ = 16 levels of ellipse orientation and 2^4^ = 16 levels of laser intensity in the photosensitive glass (vertical and horizontal axes, respectively). The measured fluorescence intensity versus the incident intensity level is displayed in Figure 2b. Fluorescence intensity at level 5 (45°) was observed to be the highest and increased almost linearly while increasing DLW intensity. Ideally, the measured fluorescence intensity should only depend on the femtosecond laser intensity during the inscription process. Experimental data from Figure 2b partially deviates from such an ideal case, since minimum fluorescence intensities for different orientation levels were distributed irregularly and since the associated curves do not exactly show the same slope. We believe that such variations are mainly due to the residual mismatch between the center of the laser beam and the center of the SLM mask for anamorphic focusing, in combination with a residual intrinsic astigmatism of the incident laser beam. Nonetheless, the linearity with the increase of DLW intensity was mostly maintained. Such a calibration matrix was further used to retrieve 5D stored data in the decoding process.

## 3. Results

### 3.1. Laser Writing Process with 256 Levels: 16 Fluorescence Intensity Levels and 16 Ellipse Orientation Levels

Images of two French Nobel Prize winners in physics were selected for simultaneous DLW using both incident intensity and ellipse orientation direction. Each image, consisting of 100 × 100 pixels, was converted to 4-bit bitmap format based on 2^4^ = 16 ellipse orientation levels (Figure 3a) and 2^4^ = 16 intensity levels (Figure 3b), respectively. Figure 3c shows the resulting entangled image at the focal plane inscribed by Type A DLW with 256 = 2^4^ × 2^4^ levels. Each ellipse pattern of 7 μm spacing was produced by irradiating 10^6^ laser pulses. The total size of the recorded image was 700 × 700 μm since the image has 100 × 100 storage units.

As shown in Figure 3c, the image recorded using laser irradiance is visually distinguished in the entangled 5D ODS image. In contrast, the image recorded by the ellipse orientation was barely visible due to the irregular fluorescent intensities for the different orientation levels, as mentioned above. Therefore, a fluorescence calibration matrix was required to retrieve the ellipse orientation of each pattern. Figure 3d displays a fluorescence calibration matrix with 16 × 16 intensity and orientation levels. The matrix was simultaneously fabricated in a separate region near the embedded DLW image in order to minimize the decoding errors.

### 3.2. Reading Process with 256 Levels: 16 Fluorescence Intensity Levels and 16 Ellipse Orientation Levels

In the readout process, we carefully adjusted the z-position of the maximal fluorescence intensity, which corresponds to the tangential focus during inscription. The field-of-view non-uniformity of the scanning confocal fluorescence microscope was also reduced by selecting a rather small area of interest (124.67 × 124.67 μm, 4096 × 4096 numerical pixels, 300 ns pixel dwell time, 4 frame average). The imaging process was repeated at different selected regions to cover the full 5D ODS image.

Each of the 100 × 100 entangled patterns of Figure 3c, namely 10 × 10 entangled patterns successively recorded in 10 × 10 selected regions, was systematically compared to the 256 patterns of the calibration matrix for decoding. Comparisons were made by calculating the root-mean-square value between the confocal image of each storage unit of Figure 3a with each pattern of the calibration matrix. Such numerical comparison was required to successively restrict both the images in Figure 3c,d to a region of interest being carefully centered in at the centroid of each of the compared patterns (each pattern of Figure 3c being compared to each of the Figure 3d patterns). By performing these automatic one-to-one comparisons using the least squares method, the most similar pattern in the calibration matrix was selected as the pattern corresponding to the minimal root-mean-square value. Note that, taking into account the fact that the reference matrix itself shows deviations from the ideal expected intensity dependence (as discussed with Figure 2b), the reported numerical decoding method is the approach that led to the best readout results with the present data, showing better fidelity than that obtained while trying to independently retrieve the intensity levels on the one hand and orientation levels on the other hand. As a consequence, the reported readout approach allowed us to determine at once the entangled orientation level and intensity level simultaneously for each of the 100 × 100 entangled image patterns. Based on such a direct decoding process, the merged laser-engraved image was successfully split into two decoded images.

The retrieved images were decoded over 16 orientation levels and 16 intensity levels, as shown in Figure 4a,b. In addition, Figure 4a,b faintly show that consecutive adjacent images were obtained after dividing the entangled image into 10 × 10 square regions. Detailed error analysis was performed to quantify the fidelity of the readout process. Comparing each of the two decoded images with the original images, it was found that the orientation and intensity readout fidelities were 61% and 25%, respectively. The orientation readout errors showed 34% 1-level error and 5% 2-level error. The intensity readout errors showed 43% 1-level error and 32% 2-level (or more) error. The readout fidelity of the intensity level was lower than that of the elliptical orientation level.

## 4. Discussion

The experimental demonstration of 5D ODS based on oriented elliptical fluorescent patterns is demonstrated. Fidelity is not yet mature to achieve transfer technology for commercial applications. However, we believe that the writing process is robust as well as the readout process while the imaging part itself showed technological limitations. Indeed, the difference in terms of orientation or intensity fidelities is mainly due to the field-of-view non-uniformity of the scanning confocal fluorescence microscope in the decoding process. A minor contribution to fidelity loss is still possible from the residual mismatch between the center of the laser beam and the center of the SLM mask for anamorphic focusing in the process of writing. A minor residual intrinsic astigmatism of the laser beam itself may also affect the inscription process and mix the orientation/intensity information supported by the storage data units and the reference patterns. Possible improvements are thus expected, especially after performing a fine intensity calibration of the field of view of the confocal microscope used for the imaging part, so as to correct sensitivity inhomogeneities. Such sensitivity inhomogeneity of the imaging system field of view affects not only the image to be decoded but also the imaging of the reference matrix itself, which can further introduce additional errors leading to the degradation of the overall process fidelity. Indeed, the non homogeneous sensitivity of the field of view of the used commercial confocal microscope introduces a quasi-random deviation in the readout and decoding process because optical data storage units are almost never observed at the same positions in the field of view for the reference image of the calibration matrix and for each of the 10 × 10 selected regions to reconstruct the full entangled image. Such critical aspects and error sources in the readout fidelity is thus considered to result mostly from the confocal imaging of the entangle image.

Beyond the readout errors that result from the experimental deviations, one can also mention potential error sources in the numerical process of pattern identification. The developed algorithm to quantify the root-mean-square error is a fully deterministic process. However, it may be sensitive to initial conditions. Indeed, for each given data storage unit to be compared to each of the 256 reference patterns, one needs to define relevant associated regions of interest in order to estimate the associated positions of the centroids. A moderate deviation of the centroid positions can affect the numerical overlapping the regions of interest to compare, which can in turn affect the least square method that drives the decision-making process of pattern identification. The estimation of such a numerical sensitivity of the decoding process would require further computation effort and readout computation time, which is beyond the scope of proof-of-concept demonstration.

The storage capacity of the proposed ODS was estimated by measuring the intensity profile of the storage unit. Figure 5a shows the fluorescence top-view image of a ODS unit measured for an area of 7.6 × 7.6 μm. Figure 5b represents the intensity cross-section profiles of basic storage units along the *y* and *x* axes. From these two profiles, we can find that at least a 5 × 5μm space is required to prevent overlapping from adjacent patterns. Figure 5c shows an intensity profile measured at 500 nm intervals on the z axis near the tangential focal plane. The fluorescence intensity depth distribution has a Gaussian distribution and shows the strongest signal at the focal plane. It indicates that a 20 μm gap is required between successive layers to prevent in-depth crosstalk. Optical data storage capacity can be estimated by taking into account the minimum volume of storage unit and 8-bits format (4-bits for intensity, 4-bits for orientation). Thus, a storage density of greater than 10 Gb/cm^3^ can be reasonably expected with the current experimental conditions.

## 5. Conclusions

We have demonstrated 5D optical data storage encoded in orientated type A DLW modifications in the silver-containing commercial glass by using a relatively low laser intensity (with very high number of pulses with sub-100 nJ pulse energies) compared to conventional DLW (that usually relies on low number of pulses at the μJ pulse energy level). Five dimensions were achieved by introducing the 3D distribution of oriented elliptically-shaped fluorescent patterns inscribed by an fs laser. The ellipse pattern was created by anamorphic focusing, and the orientation was adjusted to 16 levels by employing an SLM. In addition, an AOM device was used to adjust the fs laser intensity of 16 levels.

Two different images were embedded simultaneously at the same plane of a silver-containing commercial glass, by performing type A DLW. In particular, a 100 × 100 pixel 4-bit bitmap format image was produced for 5D recording by employing 16 ellipse orientation levels and 16 intensity levels. A calibration matrix with 16 × 16 intensity and orientation levels was also fabricated near the DLW image for the readout process. The two different original images of 4-bit bitmap format were successfully restored. The corresponding reading fidelities of 60.5% and 25.1% were obtained for the orientation direction and fluorescence intensity levels, respectively. While higher fidelities are needed while considering reliable hexadecimal information management, these results demonstrate the proof-of-concept of the proposed approach to 5-dimension multi-level ODS. Therefore, we believe that our proposed technique can be used for the next-generation optical permanent recording.

## Figures and Tables

**Figure 1 micromachines-11-01026-f001:**
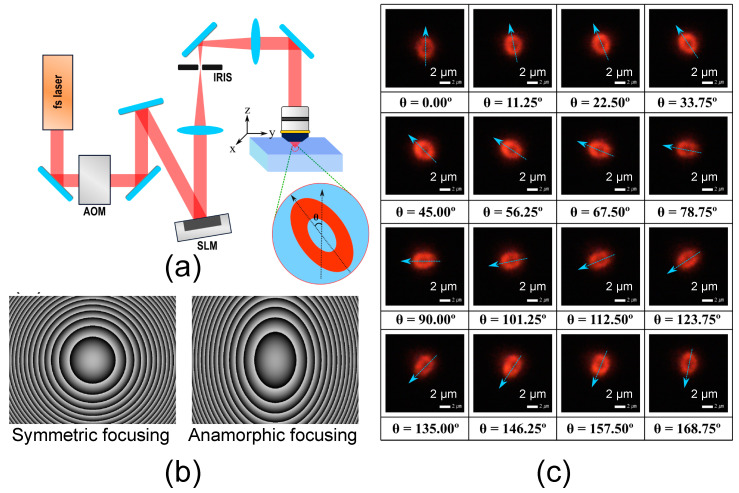
(**a**) Femtosecond laser tight focusing in the silver-containing glass, leading to the production of fluorescent silver clusters at its periphery. (**b**) SLM holographic phase masks with an additional cylindrical profile leading to an elliptical pattern by DLW. (**c**) Oriented elliptical patterns obtained by SLM phase mask manipulation, corresponding to 2^4^ = 16 orientation-encoded levels.

**Figure 2 micromachines-11-01026-f002:**
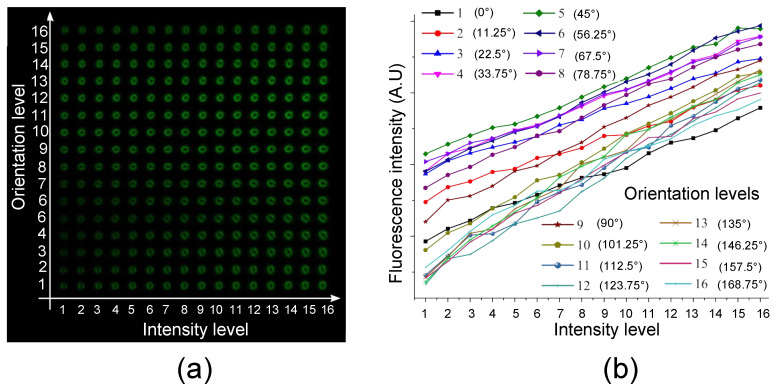
Fabricated fluorescence calibration matrix. (**a**) Confocal image of all basic storage units composed by 16 intensity levels and 16 orientation levels. (**b**) Measured fluorescence intensity versus incident DLW intensity for the 5D decoding process.

**Figure 3 micromachines-11-01026-f003:**
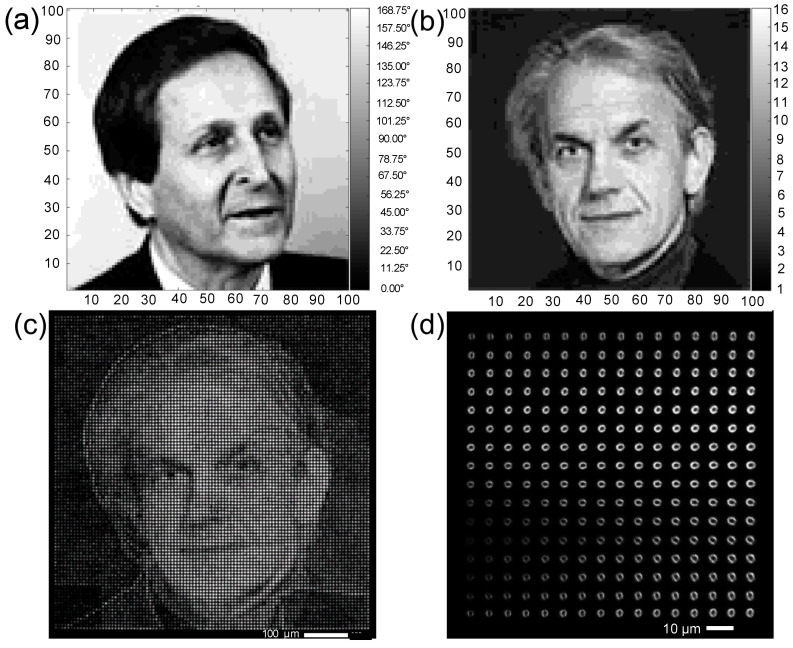
(**a**,**b**) are the encoded images of two Nobel laureates in 16 orientation levels and 16 intensity levels, respectively. (**c**) 100 × 100 entangled patterns among 16 × 16 intensity and orientation levels. (**d**) The fluorescence calibration matrix was fabricated for decoding (fluorescence excitation at 405 nm).

**Figure 4 micromachines-11-01026-f004:**
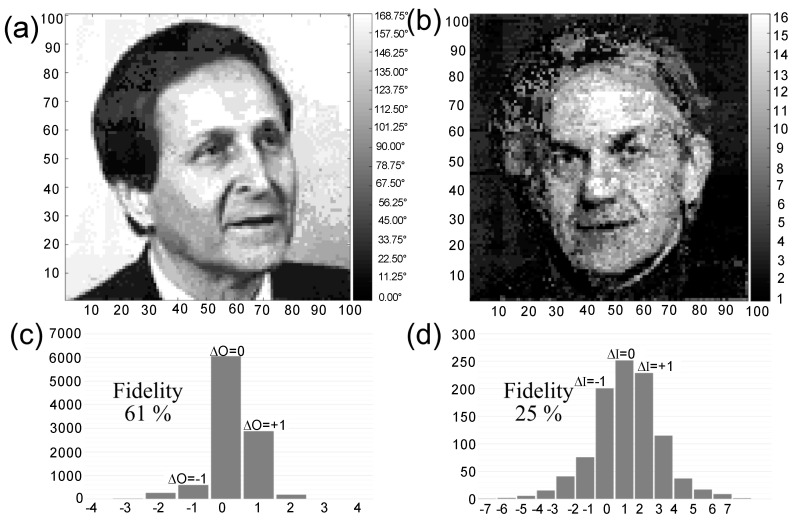
(**a**,**b**) Retrieved images from the initial images of Figure 3a,b, respectively. (**c**,**d**) Histograms of the level difference between original and decoded levels for the orientation direction and the fluorescence intensity, respectively.

**Figure 5 micromachines-11-01026-f005:**
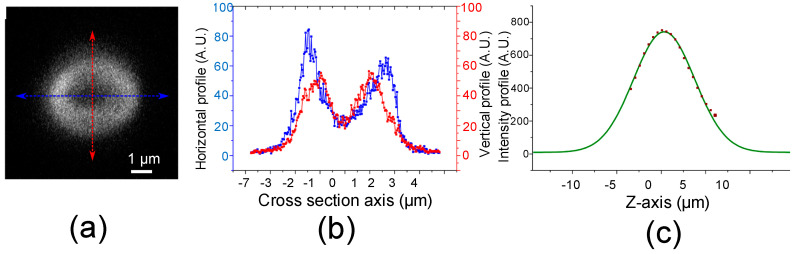
(**a**) Confocal top-view image of one single elliptically-shaped storage unit fabricated by using type A DLW. (**b**) Fluorescence intensity profile along the horizontal and vertical cross section at focal plane. (**c**) Fluorescence intensity profile and Gaussian fitting along the *z*-axis (depth).

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
