# Peer review of "Five-Dimensional Optical Data Storage Based on Ellipse Orientation and Fluorescence Intensity in a Silver-Sensitized Commercial Glass"

_micromachines, 2020, doi:10.3390/mi11121026_

Round 1
Reviewer 1 Report
This is an interesting paper for the five-dimensional optical data storage in a silver-sensitized glass. In the present study, authors have succeeded to observe a 3D distribution of oriented elliptically shaped fluorescence patterns. Overall, their experiments were performed carefully, and the paper is well organized. Thus, I would like to recommend this paper for publication just after some minor revisions.
- Page 1, Abstract: line 10, “We believe~”
Related to this sentence, how about the nondestructive emission intensity readout capability of the present system? Is the emission intensity able to keep a constant value even by the continuous light excitation? If authors have any information for the excitation time dependence (light irradiation time) on the emission intensity of the present system, it is better to explain it in the main text.
- Page 5, 3.2 title
The term “process” is duplicated.
Author Response
Dear Reviewer,
please find attached the detailed response to each of the remarks/questions.
With best regards - D. Yannick Petit

Reviewer 2 Report
This is an interesting work that focuses on creating a permanent data storage medium on surfaces. The work that was reported shows the efforts of the authors and the quality of research that was done. However, many places in the paper were somewhat ambiguous and needs clarification from the authors. The coming points are some of the issues that I found to be missing or need clarification in order to strengthen the submission.
1 - The paper states that this current approach is a 5D approach. However, these patterns were not quantified with other techniques. And in order to gain more insight into the pattern formations in such material combinations.
2- The details about the fluorescence detection techniques (system, wavelength, detector, .. etc) were not provided and it should be clear how the measurement was acquired.
3- The uncertainty sources and values should be at least mentioned.
4- The results in Figure 2-b needs to be clarified more. For example why angle (0) has high A.U. at low-intensity levels and then becomes lower than others (i.e. angle 168.75).
5- Is it possible to get an equation representing all the patterns shown in Fig 2-b, instead of reference comparisons?
6- From your point of view, what would be the best approach to identify the points where the A.U. and intensity levels match ( i.e. intensity level 7 and angles 33.75 and 67.5)?
Author Response

(The authors gave the same response as above.)
